# The Use of Flocked Swabs with a Protective Medium Increases the Recovery of Live *Brucella* spp. and DNA Detection

Luca Freddi,[a] Vitomir Djokic,[a] Fathia Petot-Bottin,[a] Guillaume Girault,[a] Ludivine Perrot,[a] Acacia Ferreira Vicente,[a] Claire Ponsart[a]

aEU/OIE & National Reference Laboratory for Animal Brucellosis, Animal Health Laboratory, Paris-Est University/ANSES, Maisons-Alfort, France

Luca Freddi and Vitomir Djokic contributed equally to this article. Author order was determined alphabetically.

**ABSTRACT** Brucellosis is a worldwide zoonosis caused by bacteria from the genus *Brucella*. Once established, it is very hard to eradicate this disease, since it contaminates animals, the environment, and humans, causing problems for veterinary and public health as well as wildlife protection programs. Swabs are used for sampling in bacteriological and/or molecular diagnostics, from seropositive animals with disease symptoms, from genitalia or tissue lesions, as well as from contaminated environments. The aim of this study was to compare main of the commercially used swab types for sampling and diagnostics of *Brucella* spp. and determine the optimal storage conditions and time frame for testing. To achieve this, we tested bacterial and molecular methods for detection of *Brucella abortus*, *Brucella melitensis*, and *Brucella suis* using nine swab types, all with different tip materials, treated immediately after spiking, after 72 h at +4°C, and after 72 h at −20°C. Flocked swabs showed the highest capacity to preserve bacterial viability and DNA quality, regardless the storage conditions. Flocked swabs immersed in a protective medium provided the best conditions for *Brucella* survival in all three storage conditions. At the same time, the efficacy of quantitative PCR (qPCR) detection for all swabs, including the positive control, was above 50%, irrespective of the storage conditions, while bacterial survival was significantly lowered when swabs were kept at +4°C or −20°C for 72 h (48.2% and 27.5%, respectively). Compared to the positive control and other types, the flocked swabs maintained higher reproducibility regarding their capacity to preserve live bacteria in all three storage conditions.

**IMPORTANCE** In order to protect public and veterinary health from highly zoonotic bacteria such as members of the genus *Brucella* and prevent their dissemination into the environment, direct diagnostics are of utmost importance. However, in addition to the highly specific diagnostic tests, the sampling methods, time necessary for specimens to reach the laboratories, and transport conditions are important factors to consider in order to increase the sensitivity of performed tests, especially bacterial culturing and qPCR. This paper shows how different swab types and storage conditions influence classical bacteriological diagnostics of the most prevalent *Brucella* species – *B. melitensis*, *B. abortus*, and *B. suis* – but have little impact on molecular methods. The presented results highlight (i) the choice of swab regarding the storage and transport conditions, (ii) the importance of immediate swab treatment upon sampling, and (iii) that molecular methods do not depend on storage conditions, unlike classical bacteriological isolation.

**KEYWORDS** swab, *Brucella*, storage condition, recovery, qPCR, bacteriology

Address correspondence to Luca Freddi, luca.freddi@anses.fr.

Brucellosis is a worldwide zoonosis affecting developing economies and causing veterinary and public health threats (1–3). Out of the 12 currently described *Brucella* species, *Brucella abortus*, *Brucella melitensis*, and *Brucella suis* are considered to be highly pathogenic to humans (4, 5). These three species of *Brucella* cause mandatorily reportable disease

throughout the world and still have high prevalence among domestic and wild animals, causing a threat to public health (6). They affect mainly cattle, small ruminants, and pigs, respectively, causing abortions and infertility (7, 8). Infected animals rarely show clinical symptoms. Those usually occur during pregnancy in females, as miscarriages, which can also be followed by long-lasting vaginal discharge and, in the worst cases, result in sterility (7). In males, *Brucella* can cause orchitis, epididymitis, and sterility and during reproduction can be disseminated within the herd. Excretion of *Brucella* is reported in sick as well as asymptomatic animals, up to $10^{13}$ CFU/ml in allantois fluids (9). Infected animals can also shed *Brucella* via milk, urine, and sperm (10). People become infected most often by consumption of unpasteurized dairy products or through direct contact with fluids from an infected animal or a contaminated environment. In humans, brucellosis is usually a chronic disease, characterized by undulant fever and lymph node enlargement. Although human brucellosis has a low mortality rate, it can be severely debilitating and disabling, resulting in endocarditis or, when it affects the central nervous system, meningitis or brain abscesses (5, 11). The tendency of the disease to present a chronic and persistent form in humans results in granulomatous disease capable of affecting any organ system (1).

In animals, *Brucella* can be isolated via collection of genital swabs after miscarriage or other clinical manifestations. To confirm the presence of *Brucella*, the European Union (EU) and World Organisation for Animal Health (OIE) consider bacterial culture as a referent method (6), while in recent years, fast and precise molecular methods have been developed (12–17). The swab type and storage temperature could influence successful bacterial isolation and molecular detection (18–22). Currently, no protective transport medium has been shown to increase bacterial survival. Moreover, freezing the swabs (except vaginal swabs) if transport lasts for more than 12 h is recommended by the OIE (6). A genital swab taken after abortion or parturition is an excellent source for the recovery of *Brucella* and is easier for laboratory personnel to handle than miscarriage material. In all cases, the shorter the shipment and storage time, the higher the probability of *Brucella* isolation, especially if the initial amount of bacteria is small or the sampled material is not fresh. Transport time is one of the most important factors in *Brucella* isolation (23). OIE guidelines instruct that field samples should be delivered to diagnostic laboratories as soon as possible (6), while the EU Reference Laboratory for Brucellosis recommends a maximum delay of 48 h between sampling and its arrival at the laboratory (European Union Reference Laboratory for Brucellosis: *Brucella* culture and genus identification - Standard Operating Procedure. https://sitesv2 .anses.fr/sites/default/files/EU%20Brucellosis%20culture%20and%20identification%20SOP% 202021%20v4.pdf). With this submission timeline, diagnostics are performed on average 3 days after sample collection. Commercially available swabs are currently used, with a large variety of tip materials (e.g., nylon, rayon, cotton, polyester, polyurethane, and alginate polymers) and microstructures (e.g., tightly wound, knitted, flocked fibers, and reticulated). The characteristics of the swab tips directly impact *Brucella* culture detection. PCR results are influenced by the amount of recovered DNA, which depends on the type of swab used as well as the DNA extraction method (23, 24).

As the methods and materials for swab production evolve, the aim of this study was to (i) compare the commercially available swab types for recovery of *Brucella* in bacterial culture and its DNA in quantitative PCR (qPCR); (ii) identify the best transportation conditions for direct detection of *Brucella*; and (iii) compare bacterial culture and qPCR for detection of *B. abortus*, *B. melitensis*, and *B. suis*. In order to achieve the objectives, we tested bacterial culture and qPCR diagnostics on nine different spiked swabs, treated immediately or after 72 h and stored under two temperature conditions.

## RESULTS

**Different swab materials have variable retention capacities.** To select the best possible representative of the materials used for the swab tip, the adsorption of all commercially available swabs was compared. Two swabs with tips made out of hydrophobic materials, with an absorption capacity of less than 50 $\mu$l, could not retain liquid and were removed from further experiments (Table 1).

Microbiology
Spectrum

**TABLE 1** Absorption properties and technical data of the swab types tested in this study

| Company name | Reference[a] | Code | Liquid absorption capacity (µl)[b] | Tip diameter (mm)[a] | Tip length (mm)[a] | Tip material[a] | Transport medium[a] |
|---|---|---|---|---|---|---|---|
| Copan | ETP | S1 | 85–90 | 4.50 | 15.00 | Viscose | No |
| | ETB | S2 | 135–155 | 4.80 | 15.00 | Cotton-wool | No |
| | ETBC | NA[c] | 0–5 | 5.00 | 15.00 | Carded cotton | No |
| | 159C | S3 | 130–140 | 4.50 | 16.00 | Polyester | No |
| | 552C (FLOQSwabs) | S4 | 175–185 | 5.5 | 16.00 | Nylon flocked | No |
| | 480CE (eSwab) | S5 | 175–185 | 5.5 | 16.00 | Nylon flocked | 1 ml Amies modified medium |
| Puritan | 25-806 1PA BT | S6 | 220–255 | 5.15 | 17.45 | Calcium alginate | No |
| | 25-1506 1PF BT | NA | 0–5 | 4.77 | 16.00 | Foam | No |
| | 25-3406-H BT (HydraFlock) | S7 | 180–185 | 5.55 | 16.25 | Polyester flocked | No |

[a]Values according to the manufacturer.
[b]Minimum and maximum absorption volume from four independent repetitions.
[c]NA, not applicable.

**Brucella species have different survival capacities on swabs and the same detection levels in qPCR.** Figure 1A shows the survival rates of the three *Brucella* species on the selected swab types and three storage conditions, analyzed together. Out of the three strains, 16M and 544 had a mean recovery rate (% recovery of live *Brucella*) from the swabs of 55.8% and 50.5%, respectively, where no significant difference was observed ($P = 0.40$) (Table S2). At the same time, strain Thomsen appeared to be more fragile: less bacteria survived the swab environments and storage conditions (mean value, 37.7%). Therefore, Thomsen had a significantly lower recovery rate compared to 16M and 544 ($P < 0.0001$ and $P = 0.005$, respectively) (Fig. 1A).

Figure 1B shows that there was no statistically significant difference in the recovery rates between the three *Brucella* species when bacteria were detected based on the

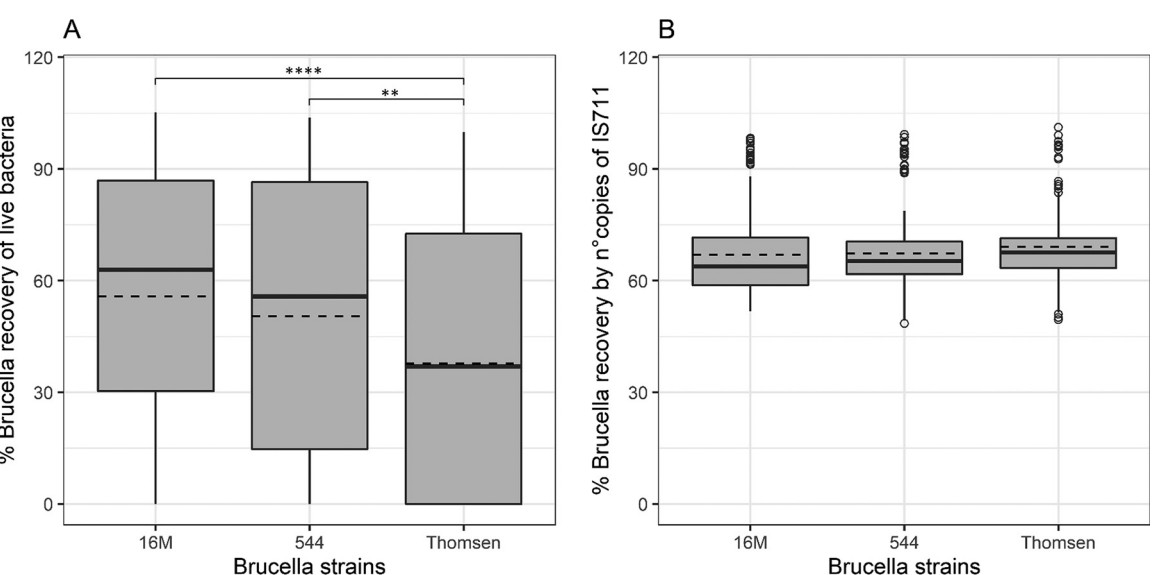

**FIG 1** Recovery rates of live bacteria (A) and their DNA (B) from three *Brucella* strains using spiked swabs. Three *Brucella* strains, representative of their species, were used: *B. melitensis* strain 16M, *B. abortus* strain 544, and *B. suis* strain Thomsen. As shown in panel A, strain Thomsen showed significantly less recovery, indicating that it is more sensitive to the swab environment under different storage conditions. There was no difference in the rate of recovery of live 16M and 544 strains when all swabs were compared. As indicated by panel B, DNA from all three *Brucella* strains was detected in the same amounts when the results of all the spiked swabs were pooled. The molecular detection method showed less variability and a higher recovery rate than bacteriological plating, without any effect from storage conditions. The solid and dashed lines represent the median and mean of the distribution, respectively. Statistical analysis was conducted using a nonparametric Tukey test with 95% confidence interval. Each bar represents the mean ± SD (standard deviation), and when the significant difference between compared values (P values) was detected, the rating was based on the following scale: *, $P < 0.05$; **, $P < 0.01$; ***, $P < 0.001$; ****, $P < 0.0001$.

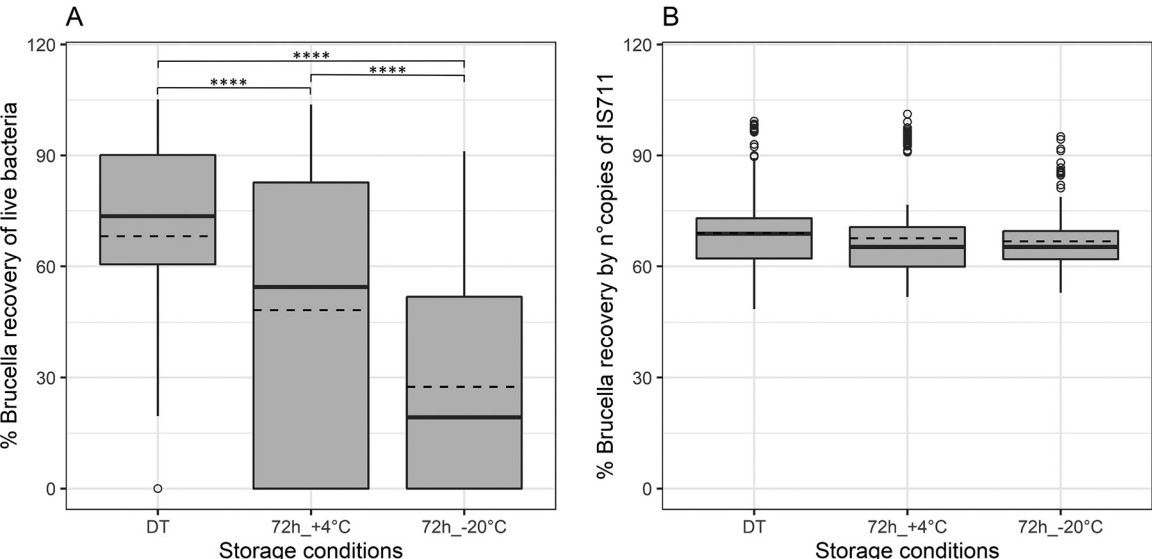

**FIG 2** Rates of detection of live bacteria (A) and DNA (B) in *Brucella*-spiked swabs stored under different temperature conditions. The swabs were subjected to three storage conditions: immediately treated (DT), 3 days (72 h) at +4°, and 3 days at –20°C. As shown in panel A, when the swabs were treated immediately, significantly larger amounts of live bacteria were recovered than after 72 h storage at +4°C and even less after freezing at –20°C. As indicated by panel B, the results show that there was no statistical difference in qPCR detection between the three storage conditions. The solid and dashed lines represent the median and mean of the distribution, respectively. Statistical analysis was conducted using a nonparametric Tukey test with 95% confidence interval. Each bar represents the mean ± SD (standard deviation), and when the significant difference between compared values (*P* values) was detected, the rating was based on the following scale: *, $P < 0.05$; **, $P < 0.01$; ***, $P < 0.001$; ****, $P < 0.0001$.

number of IS*711* copies. As expected, the variability of qPCR detection was lower between the three species, since bacterial DNA was detected regardless of the survival rates (Fig. 1B). The bacteriological recovery of all three *Brucella* strains (mean values, 55.8%, 50.5%, and 37.7% for 16M, 544, and Thomsen, respectively) was significantly lower than for molecular detection (mean values, 70.0%, 67.2%, and 69.1%, respectively) ($P < 0.0001$, compared to both conditions), especially for Thomsen (Table S2).

**Storage conditions negatively impact *Brucella* survival on swabs but not molecular detection.** The survival of live bacteria on all swab types was dependent on the storage conditions, as shown in Fig. 2A. The recovery rate of the immediately treated swabs (mean value, 68.2%) was significantly higher than that of swabs stored at +4°C or −20°C (mean values, 48.2% and 27.5%, respectively) (Table S2) ($P < 0.0001$, compared to both conditions). Still, bacteria absorbed into swabs and stored for 3 days in the refrigerator survived on average 20% longer than when frozen at −20°C ($P < 0.0001$) (Fig. 2A). Live bacteria were isolated with at least a 20% recovery rate from nearly all directly treated swabs, apart from a few repetitions of swabs S2 and S6, where no bacteria were recovered. The qPCR detection of *Brucella* did not show significant variation in quantification between storage conditions, nor between (i) immediately treated swabs and those stored at +4°C for 3 days ($P = 0.39$), (ii) swabs stored at +4°C and −20°C ($P = 0.69$), and (iii) swabs frozen at −20°C and immediately treated ones ($P = 0.08$) (Fig. 2B). Additionally, the percentages of recovered live *Brucella* and qPCR detection were similar when the swabs were treated immediately (mean values, 68.2% and 69.0%, respectively), while they differed proportionally when the swabs were stored in the refrigerator (mean values, 48.2% and 67.6%, respectively) and frozen at −20°C (mean values, 27.5% and 66.7%, respectively).

***Brucella* recovery and detection depend on different swab types regardless of the storage conditions.** With all but one of the six dry swab types tested, *Brucella* had a lower survival rate than in the control group of saline solution (PC) (mean values: PC = 71.1%, S1 = 42.1%, S2 = 10.4%, S3 = 45%, S4 = 68.7%, S6 = 17.4%, and S7 = 36.9%; Fig. 3A and Table S2). Only from the flocked dry swab (S4) were bacteria recovered at a similar rate as from saline solution without a significant difference ($P = 0.99$), when all storage conditions were included in the comparison (Fig. 3A). Interestingly, the only wet swab tested (also flocked – S5) provided the best conditions for *Brucella* survival under all three storage

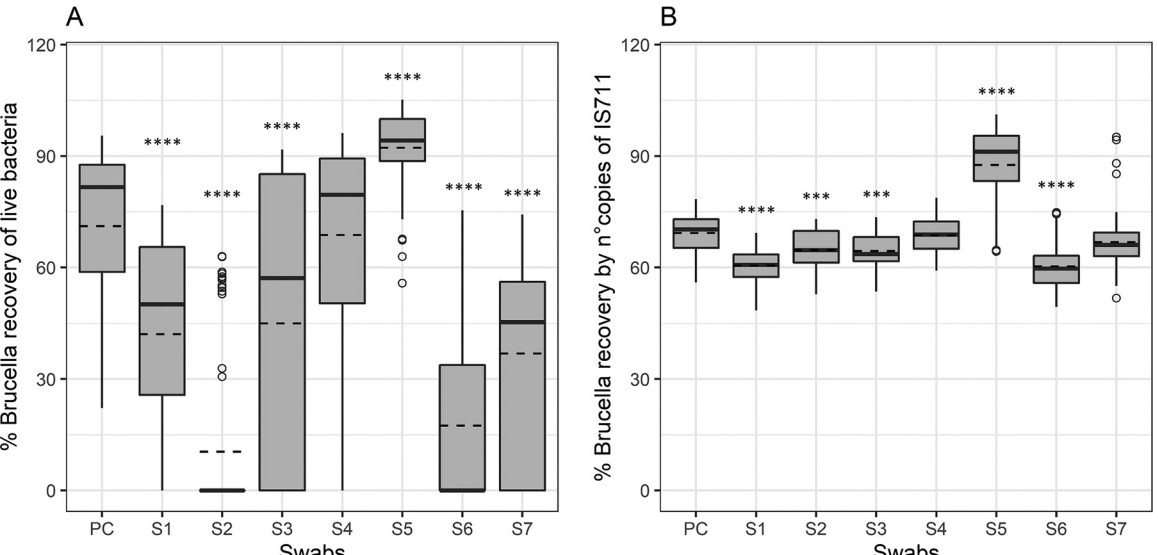

**FIG 3** Rates of detection of live *Brucella* cells and DNA by bacterial culture (A) and qPCR (B), respectively, from spiked swabs under all three treatment conditions together. Recovery of live bacteria and detection by qPCR were compared on seven swabs and a positive control (PC): S1, viscose; S2, cotton-wool; S3, polyester; S4, nylon flocked; S5, nylon flocked with medium; S6, Ca-alginate, and S7, polyester flocked. As shown in panel A, the recovery rates from different spiked swabs depended on the type of materials used for the tip. The flocked wet swab (S5) showed the highest recovery rate, significantly higher from all others and the PC. Only from the dry flocked swab (S4) was the same amount of bacteria recovered as from the PC, while all other materials showed significantly less bacterial protection. As indicated by panel B, the rates of qPCR detection of *Brucella* DNA were more constant throughout the course of the experiment. Still, S5 released more DNA, which was easier to detect than from other swab types. From the two dry flocked swabs (S4 and S7), the same ratio of DNA was detected as from the PC, while from the other swab types, significantly less bacterial DNA was detected, negatively affecting the qPCR. The solid and dashed lines represent the median and mean of the distribution, respectively. Statistical analysis was conducted using a nonparametric Tukey test with 95% confidence interval. Each bar represents the mean ± SD (standard deviation), and when the significant difference between compared values ($P$ values) was detected, the rating was based on the following scale: *, $P < 0.05$; **, $P < 0.01$; ***, $P < 0.001$; ****, $P < 0.0001$.

conditions (mean value, 92.2%) and was significantly different from the control ($P < 0.0001$). The cotton-wool swab (S2) provided very little to no protection of the bacteria compared to the control ($P < 0.0001$), although the fiber is similar to that of carded cotton (S3) (Fig. 3A).

At the same time, the efficacy of qPCR detection for all swabs as well as the positive control was above 48% (mean values: PC = 69.3%, S1 = 60.5%, S2 = 64.7%, S3 = 64.4%, S4 = 68.6%, S5 = 87.6%, S6 = 60.3%, and S7 = 66.8%) (Fig. 3B and Table S2). Equally intriguing is the fact that significantly less bacteria was identified from the dry, non-flocked swabs than from the positive control (Fig. 3B). Although the percentages of live bacteria isolated were significantly different between the two dry flocked swabs (S4 and S7; $P < 0.0001$), both were equally effective for detection of *Brucella* DNA ($P = 0.73$). Compared to the dry swabs, the pattern of live bacteria isolation from the wet swab (S5) showed that a significantly higher number of IS*711* copies was detected ($P < 0.0001$). Interestingly, the mean numbers of bacteria isolated from the two nylon flocked swabs (S4 and S5) and the positive control (mean values, 68.7%, 92.2%, and 71.1%, respectively) were higher than the rate of detection of IS*711* copies by qPCR from the same swab types and control (mean values, 68.6%, 87.6%, and 69.3%, respectively) (Fig. 3A and B and Table S2). On the other hand, in S1, S2, S3, S6, and S7, DNA was more easily detected using molecular methods than were live bacteria.

***Brucella* survival is highly dependent on the type of swab used for its isolation, while molecular detection is more uniform irrespective of the storage conditions.** From dry swabs S1, S3, S4, and S7, the mean recovery of live *Brucella* was well above 55% when they were treated immediately after spiking (mean values, 66.1%, 85.2%, 88.2%, and 55.8%, respectively), but for all four of these swabs, the percentages decreased significantly when stored for 3 days at +4°C (mean values, 28.7%, 48.8%, 81.2%, and 48.2%, respectively) (Table S2), and even more so after freezing (mean values, 31%, 1%, 36.8%, and 6.7%, respectively) (Fig. 4A). However, compared to the positive control, S4 and S5 maintained higher reproducibility regarding their capacity to

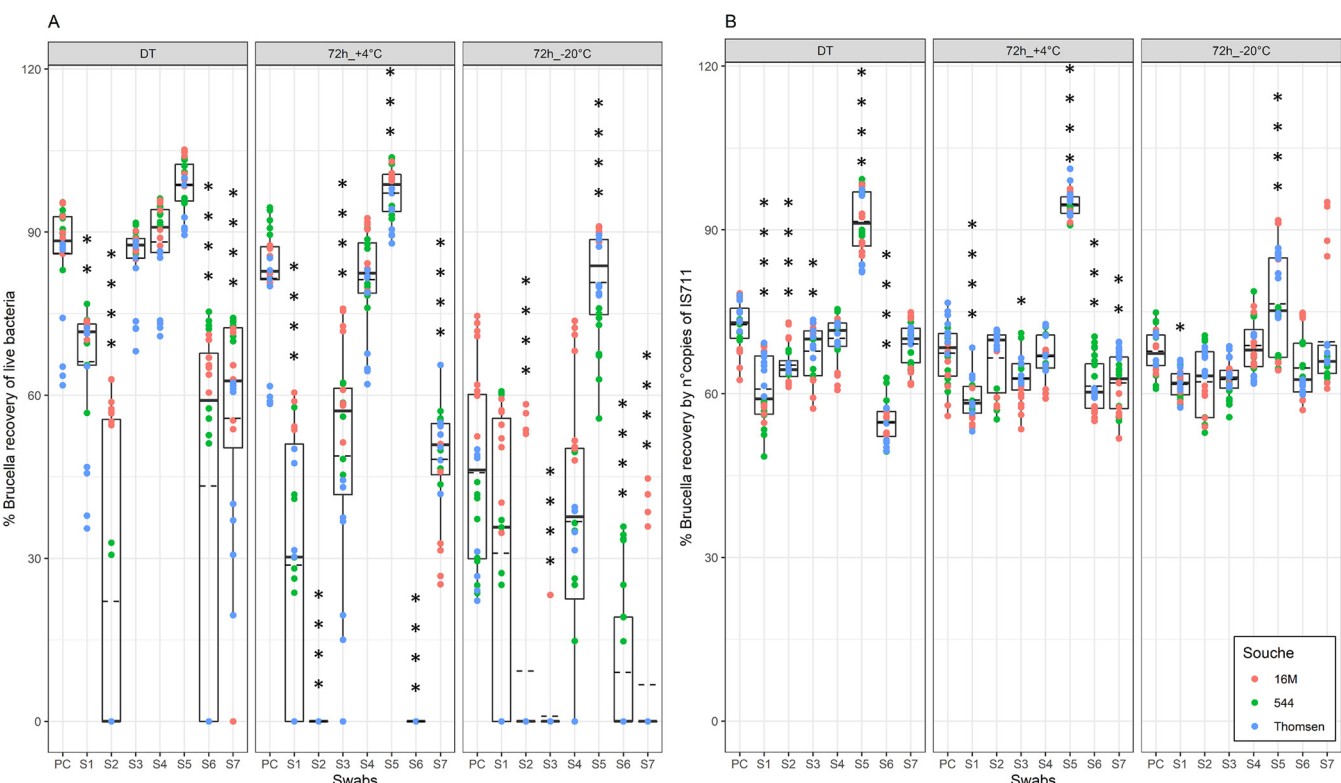

**FIG 4** Live *Brucella* (A) and their DNA (B) recovery rates from seven swab types compared to positive control (PC), under three storage conditions. The seven swabs were as follows: S1, viscose; S2, cotton-wool; S3, polyester; S4, nylon flocked; S5, nylon flocked with medium; S6, Ca-alginate, and S7, polyester flocked. The swabs were subjected to three storage conditions: immediate treatment (DT), 3 days (72 h) at +4°, and 3 days at −20°C. As indicated in panel A, the rate of detection of live *Brucella* cells depended heavily on the type of swab used in all three storage conditions. When the swabs were immediately treated (DT), from four swabs (S1, S2, S6, and S7), significantly less bacteria was detected than from the PC, while only from S4 and S5 was more bacteria isolated than from the control. In some cases (from S2, S6, and S7), no CFUs were isolated. When swabs were stored for 72 h at +4°C, only the use of S5 (wet flocked) led to the isolation of considerably more bacteria. From S1, S2, S3, S6, and S7, significantly less, and in some cases no, bacteria was isolated compared to the PC. When the swabs were stored for 72 h at −20°C, only from S5 spiked swabs live bacteria were isolated, regardless of the *Brucella* strain used. Only S4 showed the same recovery rate as the PC under all storage conditions. As shown in panel B, the storage conditions had no significant effect on the DNA detection levels. Only from S5 was significantly more bacterial DNA detected, while the other two flocked swabs (S4 and S7) showed no difference in *Brucella* molecular detection, compared to the PC in all three storage conditions. While the qPCR detection varied with swab type for DT and storage at +4°C compared to the PC, this variation was less significant when the swabs were frozen at −20°C for 3 days. The solid and dashed lines represent the median and mean of the distribution, respectively. Statistical analysis was conducted using a nonparametric Tukey test with 95% confidence interval. Each bar represents the mean ± SD (standard deviation), and when the significant difference between compared values (P values) was detected, the rating was based on the following scale: *, $P < 0.05$; **, $P < 0.01$; ***, $P < 0.001$; ****, $P < 0.0001$.

preserve live bacteria in all three storage conditions (mean values, 88.2% and 98.7% for immediate processing after spiking (DT), 81.2% and 97.2% at +4°C for 72 h, and 36.8% and 80.7% at −20°C for 72 h, respectively) (Table S2). For S2, S6, and S7, even if treated immediately, the bacterial survival rate was significantly lower than that of the positive control ($P < 0.0001$) and dropped below 10% when the swabs were frozen (mean values, 9.2%, 9%, and 6.7%, respectively) (Fig. 4A). Only from the wet swab (S5) were all tested *Brucella* strains rescued from each repetition, in all three storage conditions.

Regardless of the storage conditions, qPCR was able to detect *Brucella* DNA from all swab types with a minimal recovery rate higher than 54% (Fig. 4B). However, the maximal level of detection stayed on average below 76% in all storage conditions, except for S5 (mean values, 91.5%, 94.7%, and 76.5% at DT, at +4°C for 72 h, and at −20°C for 72 h, respectively) (Table S2). All dry nonflocked swabs had a lower detection rate by qPCR immediately after being treated compared to the control (mean values, PC = 72.6%, S1 = 60.8%, S2 = 65.3%, S3 = 67.8%, and S6 = 54.7%) (Fig. 4B). Moreover, only for swab S1, significantly fewer bacteria (mean values, PC = 72.6% and S1 = 60.8% at DT, PC = 67.5% and S1 = 58.8% at +4°C for 72 h, and PC = 67.7% and S1 = 61.7% at −20°C for 72 h; $P < 0.022$) were detected than from the positive control in all tested conditions (Fig. 4B). Only the wet swab (S5) maintained an efficacy of above 64% in all

three tested conditions with all three species of *Brucella* used and detected significantly higher numbers of bacteria than for the positive control ($P < 0.0001$).

## DISCUSSION

Typically, when swabs are used for sampling after miscarriage for *Brucella* detection, they need to be transported to the laboratory immediately and inoculated onto solid medium. This may take several days, depending on the geographical distance from the laboratory or the transportation speed. The OIE manual advises against freezing genital swabs during transport to preserve sample integrity (6). However, no guidance is given concerning the swab type or whether a transport medium should be used, in order to ensure high sensitivity of the *Brucella* detection assay. This lack of information may be overcome by the fact that when there is a suspicion of *Brucella* as the causative agent of miscarriage, bacteria can be excreted in billions or trillions (up to $10^{13}$ bacteria) for a short period and potentially easily detected with all types of swabs (9). However, when the animal is serologically positive but without specific clinical manifestations (fever, cutaneous wounds, orchitis, epididymitis, lymphadenitis, etc.) or for environmental swabs, the number of suspected bacteria could be far lower, making detection more challenging. For this reason, we investigated the impact of various commercially available swabs (S1, S2, S3, S4, S5, S6, and S7) and three storage conditions (immediate diagnostics, after storage for 72 h at $+4°C$, and after storage for 72 h at $-20°C$) on the rates of detection and recovery of three prominent *Brucella* species worldwide (16M, 544, and Thomsen).

There is no doubt that swab type plays a major role in bacterial preservation and recovery, but it has no significant effect on molecular detection. Our results are concordant with the OIE recommendation not to freeze the swabs in order to increase the chance of live *Brucella* isolation. Storing the *Brucella*-spiked swabs at $-20°C$ for 3 days resulted in a significant decrease in the recovery rate by bacterial culture, while the molecular method maintained the same level of detectability (Fig. 2A and B). Additionally, even swab storage at $+4°C$ decreased the percentage of recovered bacteria (Fig. 2A). The freeze-thawing of samples is thought to potentially destroy target DNA and affect the sensitivity of the PCR assay (25). However, over the last decade, advancements in sample preservation have resulted in improved transport, ensuring constant conditions (e.g., temperature, humidity, and time) and their traceability. Additionally, DNA extraction and molecular diagnostic techniques have improved the detection sensitivities from frozen matrices, especially when used in combination. For example, from spiked clinical specimens (including swabs), *Chlamydia* DNA detection is not affected by different storage temperatures ($4°C$, $-20°C$, and $-80°C$) except after prolonged periods (more than 2 years) (20). Moreover, our results highlight that qPCR detection is less affected than isolation of viable bacteria from frozen swabs. While *Brucella* start to deteriorate in swabs after 3 days at $+4°C$ and $-20°C$, molecular detection rates at those times are equivalent to those for immediately treated swabs. Refrigeration has been shown to increase the efficiency of *Mycoplasma* detection in swabs compared to room temperature storage, suggesting that transporting samples on ice is advantageous for detection (18, 26). However, the impact of temperature on sample preservation must be considered in its entirety. Our results, with the use of both wet and dry swabs, show very low survival rates for *Brucella* even in cooling conditions at $+4°C$, and even lower to almost none when frozen at $-20°C$. While the advantages of freezing samples can be questioned, it remains one of the most common approaches to preventing changes in the abundance of specific taxa in multicontaminated samples (27–29). Consequently, freezing could remain a valid solution for the preservation of swabs (e.g., genital and rectal) intended for detection of multiple pathogens and especially for microbiome analysis. However, in that case molecular detection proved more sensitive than classical bacteriological isolation methods for diagnostics of *Brucella* species.

In our study, the number of IS*711* copies for an equal amount of bacteria was always in agreement with the results published by Dauphin et al. and Kang et al. (30, 31). Nevertheless,

the recovery rate of live bacteria obtained in this work is around 25% higher than in the results shown by Kaden et al. (23), compared with immediately treated spiked swabs. This can be explained by the differences in spiking method; in this study, the bacteria were directly dropped by pipette onto the dry head of the swab, while Y. S. Kang et al. (31) used swabs that were already inoculated with bovine vaginal excretes, therefore lowering the adsorption capacity for spiked liquids. Equally important for diagnostics, our protocol subjected the spiked swabs to overnight immersion in phosphate-buffered saline (PBS) at 37°C with two steps of vortexing to discharge as many bacteria as possible, as described in Materials and Methods.

The present results support the concept that qPCR is an even more reliable method for the rapid detection and quantification of *Brucella* in swabs than bacterial culture on solid media. The percentage of rescued bacteria compared with that detected by qPCR is comparable when the swabs were immediately treated (Fig. 2A and B). The higher mortality of *Brucella* spp. under refrigeration (+4°C) and freezing (−20°C) conditions results in better detection by molecular methods than by classical bacteriological methods. As expected, the recovery of live bacteria can vary within the *Brucella* genus, with the *B. suis* strain Thomsen less detectable than the other strains (Fig. 1A) in a bacterial culture. In fact, it is commonly accepted dogma that Thomsen is a *Brucella* strain less resistant to storage, especially under freezing conditions. From our experience, besides lower detectability in selective and nonselective media, strain Thomsen grows more slowly and takes several days to reach visible CFUs.

The use of qPCR for *Brucella* diagnosis can be an advantage because it enables the detection of live and dead bacteria while decreasing the diagnostic time and risk of infection for laboratory staff compared to culture. Furthermore, implementation of automated processing methods for DNA extraction can shorten the processing time, increase throughput capacity, and decrease the probability of human error, as well as the cost of diagnostics (30, 32).

A variety of swabs are commercially available, differing in tip material and having various chemical and physical characteristics that could affect sample collection (size of the tip, absorption capacity, shape of the tip, whether the tip dries quickly and causes dehydration of the bacteria, etc.) and release (density and charge of the fibers, presence of chemicals, etc.). Our study shows that the choice of swab significantly impacts *Brucella* detection (Fig. 3 and 4). The wet flocked eSwab (S5) showed the best level of detection using both culture and qPCR, irrespective of the storage conditions (Fig. 3 and 4, Table S1). In addition, the eSwab includes conservation medium, which protects the viability of both aerobic and anaerobic bacteria during room temperature and refrigerated storage for up to 48 h (21, 22). This goes to show that the use of protective medium, even nonselective ones, improves bacterial survival and is especially important when samples will be stored or transported for longer periods of time. Our experiments show that the use of eSwab allows longer viability of *Brucella* organisms with 3 days' storage at different temperatures, which significantly increases the diagnostic sensitivity compared to the controls and all other tested swabs (Fig. 4). The presence of Amies modified medium seems to play a key role in the protection of bacteria and DNA compared to PBS only or dry swabs. Although the precise composition is protected, the presence of emulsifier (Tween 80) in the medium could envelope bacteria and preserve their viability, especially in an environment with mucus (33, 34). However, its effectiveness could be weakened in a naturally infected sample due to the presence of *Brucella* within the host cell, acting as a natural carrier. In our experiments, only one type of swab with a protective medium was tested, and further studies should be performed to compare the usefulness of other available media and bacterial preservatives. Of the dry swabs, the nylon flocked (S4) is the only swab which did not show a significant decrease in the percentage of live bacteria recovered compared to the positive control. These flocked swabs (S4 and S5) contain long nylon fibers embedded in a hydrophilic layer, which provides better capillary action and stronger hydraulic uptake of liquids. The new tip design may increase the absorption of targeted cells by 2- to 3-fold over traditional rayon-tipped swabs and improves diagnostic sensitivity

(35). The use of flocked swabs was also reported to enhance the efficiency of bacterium recovery and DNA detection compared to traditional fibers (36, 37). Unexpectedly, another dry flocked swab (S7) tested in this study is less suitable for the protection of bacteria, while it still shows the same retention of DNA for molecular detection compared to the positive control. The HydraFlock (S7) system is an alternative flocked fiber specifically designed for the collection and release of biological specimens and provides higher absorption compared to standard nylon flock swabs. However, in our study, the variability in fiber length was a negative point, as *Brucella* spp. got stuck among the long fibers and could not be easily released.

Absorption is considered one of the most important characteristics of sampling swabs and is directly related to the absolute amount of analyte that swab can incorporate (33, 34). The higher the absorption value, the more bacteria can potentially be collected, increasing the total DNA yield or number of CFUs. As expected, our data show a higher absorption capacity for these flocked swabs (S4, S5, and S7) compared to other cotton fiber ones, but still less than for the Ca-alginate swab (S6; Table 1). The total amount of absorbed liquid is similar in both types of flocked swabs and is comparable with data reported in the literature (35, 36). These data should ultimately be compared with matrices usually collected in the field, in order to estimate the variation in the swabs' absorption capacities. Although the Ca-alginate swab (S6) has higher absorption capacities, our tests demonstrated that this characteristic alone is not crucial for bacterial protection, as is the case with sponges (37). Swabs with a larger volume capacity can collect more bacteria when an excess of fluid sample is available. Therefore, to examine purely bacterial survival and capacity to retain the DNA, in this work, all swabs were spiked with the same volume of liquid, containing an equal number of bacteria. It would also be interesting to identify which swabs provide the best bacterial survival and DNA retention at their lowest absorbance level, mimicking a situation with a limited amount of sample.

The various swab materials or sterilization processes using PCR inhibitors may be limiting factors (38–40). Our study analyzed only swabs spiked with bacteria in PBS and therefore eliminated the possibilities of tissue, environmental, or chemical inhibition of PCR. However, to test the swab tip materials as well as sterilization methods as a source of PCR inhibitors, we used the commercially available internal process control (IPC) Diagenode as the internal control for both DNA extraction and qPCR amplification efficacy. The IPC threshold cycle ($C_T$) values from internal controls were compared with tested samples for each PCR run separately. Considering that the average IPC $C_T$ differences between the controls and samples did not exceed three cycles, we concluded that there was no PCR inhibition of the tested swabs (Fig. S1).

In conclusion, to protect public and veterinary health from highly zoonotic bacteria from the genus *Brucella* and prevent spillover into the environment, the choice of appropriate sampling strategy and therefore swab is of utmost importance. Our work, presented in this paper, shows how different swab types and storage conditions influence classical bacteriological diagnostics of the three worldwide species *B. melitensis*, *B. abortus*, and *B. suis* but have little to no impact on molecular methods. The presented results show that diagnostic sensitivity increases significantly if swabs are treated as soon as possible upon sampling and that molecular methods are less dependent on storage conditions, compared to traditional methods to ensure bacterial survival and isolation.

## MATERIALS AND METHODS

**Biosafety procedures.** All procedures using virulent *Brucella* strains were performed in a biosafety level 3 laboratory. Culturing of bacteria and their inactivation for DNA extraction were conducted in a class II biological safety cabinet (Thermo Fisher Scientific, USA). Worker protection was also ensured by the use of a powered air-purifying respirator and personal protective laboratory equipment.

**Bacterial strains and culture.** Three reference *Brucella* strains were used in this study, *B. melitensis* bv. 1 strain 16M, *B. abortus* bv. 1 strain 544, and *B. suis* bv. 2 strain Thomsen, originating from stocks regularly maintained at the Department of Bacterial Zoonosis (Animal Health Laboratory, ANSES). Cultures were initiated from frozen stocks into blood agar base (Thermo Fisher Scientific, France) plates with 5% horse serum and incubated at 37°C under 5% $CO_2$ for 4 days. For each strain, a single colony was inoculated into 10 ml tryptic soy broth (TSB; bioMérieux, France) and cultured in a 5% $CO_2$ atmosphere with shaking (150 rpm) for 3 days to reach a concentration of at least $10^9$ CFU/ml. For each *Brucella* culture, a

10-fold serial dilution in phosphate buffer saline (PBS) was made and inoculated into blood agar base plates in triplicate, to estimate the exact concentration by counting the CFUs.

**Swab selection.** Among commercially available swabs, only swabs with regular-sized tips (~5 mm wide and ~16 mm long) commonly used for sampling were selected. Only swabs with a transport tube were selected, without considering the different types of handles. Furthermore, for the same type of tip material, only one representative was selected, belonging to the main brands on the market. Eight different swabs were finally chosen based on the different materials: viscose, cotton-wool, carded cotton, polyester, and nylon flocked (Copan Italia S.p.A., Brescia, Italy) and calcium alginate, foam, and polyester flocked (Puritan Medical Products Co., LLC, Guilford, ME, USA) (Table 1). One flocked eSwab (wet swab) with the same type of tip as the dry one (nylon flocked) containing 1 ml modified liquid Amies medium (Copan Italia S.p.A.) was also included in the study.

**Swab absorbance testing.** The absorbance of each swab type was calculated by adding 5 $\mu$l PBS until full capacity of the tip was reached, evidenced by dripping from the swab. This test was repeated in quadruplicate for each swab type.

**Experimental design.** Three *Brucella* cultures (16M, 544, and Thomsen) were diluted in PBS, to achieve an average concentration of 2.75 ($\pm$0.71) $\times$ $10^6$ CFU/50 $\mu$l, 5.63 ($\pm$0.57) $\times$ $10^6$ CFU/50 $\mu$l, and 4.25 ($\pm$0.35) $\times$ $10^6$ CFU/50 $\mu$l, respectively. Swabs were spiked individually with 50 $\mu$l, using a micropipette in order to ensure the exact volume absorbed. The same amount of bacteria was also used to spike 2 ml PBS as a positive control (PC) (50 $\mu$l bacteria + 1,950 $\mu$l PBS), and 2 ml nonspiked PBS was used as a negative control. Three storage conditions were selected to compare the swabs, PCs, and negative controls: immediate processing after spiking (DT), storage for 3 days (72 h) at +4°C, and storage for 3 days at −20°C. For all storage conditions and for each *Brucella* strain used, four swabs per type as well as four PCs were spiked. In order to avoid the biological effect of the swab material and variations of bacterial pipetting, two independent analyses were performed by the same person for each individual swab, accounting for a total of eight analyses per swab type, bacterial strain, and storage condition. To compare culture and molecular analysis, the spiked swab tips were immersed in 2 ml PBS, vortexed for 30 s, and incubated at 37°C overnight to discharge the bacteria. For the eSwab, only 1 ml PBS was added, because it already contains 1 ml transport medium. The PCs were only incubated at 37°C overnight, without additional PBS. All swab tips were then removed after the vortexing step, and the working suspension (WS) was used for bacterial plate culturing and DNA extraction.

***Brucella* culture recovery.** A 10-fold serial dilution of WS from $10^0$ to $10^{-4}$ were made in PBS. For all swabs, 100 $\mu$l of each dilution was inoculated onto Farrell agar plates and incubated at 37°C under 5% $CO_2$ for 4 days. All dilutions were inoculated in duplicate. The number of colonies obtained were used to calculate the quantity of bacteria recovered for each swab. The percentage of live *Brucella* recovered was calculated using the following formula: [$\log_{10}$ (number of recovered CFUs)/$\log_{10}$ (number of spiked CFUs)] $\times$ 100. When no bacteria were recovered, the recovered percentage of available *Brucella* bacteria was considered zero.

**Molecular detection of *Brucella*.** Two *Brucella* cultures per strain (16M, 544, and Thomsen) were adjusted in PBS to 1 $\times$ $10^9$ CFU/ml and 10-fold diluted to a final concentration of 1 $\times$ $10^1$ CFU/ml. DNA was then extracted in duplicate from each 10-fold dilution and qPCR performed to determine the correlation between the $C_T$ values of IS*711* and the number of CFUs for each *Brucella* species.

DNA was extracted from 200 $\mu$l WS using the High Pure PCR template kit (Roche Diagnostics, France) according to the manufacturer's instructions. Prior to lysis, 10 $\mu$l of the internal process control (IPC; DNA extraction and PCR inhibition control, DiaControlDNA, Diagenode, Liege, Belgium) was added at a concentration of 1,000 50% tissue culture infective dose ($TCID_{50}$)/ml viral load. The PCR target sequence for *Brucella* was the IS*711* intergenic spacer gene fragment present in all *Brucella* species (24). The genomes of *B. abortus*, *B. melitensis*, and *B. suis* have 7 copies of IS*711*, of which one is truncated only in *B. abortus* (41, 42). The qPCR was performed as previously described by Bounaadja et al. (43). The CFU quantity was then determined based on the calibration curve corresponding to the appropriate *Brucella* strain used for spiking. The percentage of recovered *Brucella* DNA by qPCR (based on IS711 copies number) was calculated using the following formula: [$\log_{10}$ (number of CFUs detected using qPCR)/$\log_{10}$ (number of spiked CFUs)] $\times$ 100.

**Statistical analysis.** Data were recorded and descriptive statistics calculated using R Studio (44), correlated with R version 4.0.5 (45). Multivariate statistical analyses were applied to assess the impact of the *Brucella* strain, storage conditions (temperature and time), and swab type on two dependent variables: percentage of *Brucella* CFU recovery and percentage *Brucella* recovery using qPCR (percentage of estimated CFUs by number of IS*711* copies). To compare the differences between the swabs in the described variables, analysis of variance (ANOVA) was used. Further, to identify the differences between individual variables, a *post hoc* analysis was performed using the Tukey honestly significant difference (HSD) function with a 95% confidence interval (95% CI). To analyze the mean differences between individual strain detections, the one-way Student *t* test was used with 95% CI. The *P* value was considered significant when the estimated error was less than 5% (0.05).

## SUPPLEMENTAL MATERIAL

Supplemental material is available online only.
**SUPPLEMENTAL FILE 1**, PDF file, 1.4 MB.

## ACKNOWLEDGMENTS

We thank the technical support team for medium preparation and lab management and the scientists working at the Bacterial Zoonoses Unit for critical and supportive comments.

This work was supported by the European Union Reference Laboratory for Brucellosis (grant agreement no. SI2.801943) and the One Health EJP project (JRP-17 – IDEMBRU: Identification of emerging *Brucella* species: new threats for human and animals; grant agreement no. 773830).

We declare no conflict of interest.

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
