## [Reviewer comments · Microbiology Spectrum]

Microbiology Spectrum

The use of flocked swab with protective media increases the recovery of live *Brucella* spp. and DNA detection

Luca Freddi, Vitomir Djokic, Fathia Petot-Bottin, Guillaume Girault, Ludivine Perrot, Acacia Ferreira Vicente, and Claire Ponsart

Corresponding Author(s): Luca Freddi, ANSES

Review Timeline:

Submission Date:	June 28, 2021
Editorial Decision:	July 30, 2021
Revision Received:	September 16, 2021
Accepted:	September 30, 2021

Editor: William Lainhart

Reviewer(s): Disclosure of reviewer identity is with reference to reviewer comments included in decision letter(s). The following individuals involved in review of your submission have agreed to reveal their identity: Kimberly Lehman (Reviewer #2)

Transaction Report:

DOI: <https://doi.org/10.1128/Spectrum.00728-21>

July 30, 2021

Dr. Luca Freddi
ANSES
Animal Health Laboratory
14 Rue Pierre et Marie Curie
Maisons-Alfort 94700
France

Re: Spectrum00728-21 (The use of flocked swab with protective media increases the recovery of live *Brucella* spp. and DNA detection)

Dear Dr. Luca Freddi:

Thank you for submitting your manuscript to Microbiology Spectrum. When submitting the revised version of your paper, please provide (1) point-by-point responses to the issues raised by the reviewers as file type "Response to Reviewers," not in your cover letter, and (2) a PDF file that indicates the changes from the original submission (by highlighting or underlining the changes) as file type "Marked Up Manuscript - For Review Only". Please use this link to submit your revised manuscript - we strongly recommend that you submit your paper within the next 60 days or reach out to me. Detailed information on submitting your revised paper are below.

Link Not Available

Sincerely,

William Lainhart

Journals Department
Reviewer comments:

Reviewer #1 (Comments for the Author):

In this study, Freddi et al examine the impact of different commercially available swabs on *Brucella* culture and PCR detection. It highlights the importance of collection media and storage conditions in diagnostics. My comments are as follows:

Major comments:

1. The entire manuscript needs to proofread for grammar and sentence structures, especially the Abstract, Importance, and Introduction sections
2. The swabs were assessed by inoculating them with ~106 Cfu/50 uL of bacteria. Why was this the appropriate Cfu to assess these swabs?
3. The authors conclude that the flocked swabs with protective media enhance the recovery of *Brucella* in culture and for molecular detection. This study would be of a greater impact if the authors could collect some prospective data from the field to support their observations.

Minor comments

None

Reviewer #2 (Comments for the Author):

Overall Comments:

- The paper was an interesting read and I appreciated the opportunity to review it!
- Overall, it is well done, but there are a few areas/phrases that seem awkward that I would recommend changing for clarity (described in detail below).
- It would be nice to have a little more discussion on why the 3 Brucella strains were selected somewhere in the paper. You give a brief glimpse into why the Thomsen strain was selected with the discussion about its lower resistance to storage, but more discussion about the other strains would be appreciated.
- The positive and negative controls are discussed in the experimental design section, but only discussed so far as what they are. Since a lot of your experiment and data is comparison to the positive control, I would also recommend detailing how the positive and negative controls were tested. Was it placed on a control swab and tested as well? Was the PBS solution directly tested? Were the controls tested in quadruplicate as the swabs were? Clarification and more discussion on this would be useful.
- Along those lines, the data shows that one swab type (S5) produced more live bacteria and DNA when compared to the positive control, and is pointed out several times, but there is no discussion as to how this would be possible. This should be something to clarify and discuss further.
- I would like to see more data from your testing. I would recommend including a table of results showing the calculated percent recovery of live Brucella and the calculated percent recovery of Brucella DNA. Also, it would be nice to visually see the correlation of Ct values and number of CFU's for each Brucella species.
- More numbers and data need to be given in the Results section. The tables and data shown are more comparisons of the values, but not the values themselves so there is nothing for the reader to assess except your statements about the data.
- Was any method or corrections used in the statistical analysis for repeat testing as the swabs/testing was performed in quadruplicate? This needs to be addressed.
- Daub and swab seem to be used interchangeably throughout the paper but can be confusing. I would recommend only using a single terminology consistently.
- Use of duplicate/triplicate/quadruplicate should be singular, not plural as stated multiple times within the paper.

Specific Comments:

- Line 24, remove "both" as you list more than 2 areas where Brucella can be persistent
- Line 26, "Therefore" seems to be an awkward transition phrase here. Recommend re-writing or adding another sentence to connect the thoughts from this sentence to the one before it.
- Line 27, remove "the"
- Line 29-30, "...optimal storage conditions for shipment within 72 hours." I believe you mean to say optimal storage conditions and timeframe for testing. The way this is phrased is awkward and seems to mean anytime within those 72 hours, but your experiments were either immediate or 3 days with no in-between.
- Line 32, "...after 72h at +4oC or at -20oC." I would recommend re-writing to something like "after 72h at +4oC and after 72h at -20oC." As written, it seems like the swabs were +4oC or -20oC.
- Line 32, remove "That"
- Line 34, make "swabs" singular as there was only 1 tested with protective media
- Line 35, 55% is listed but there's no reference to the reader on how that compares to the other swabs to know if 55% is good or bad. I'd recommend adding additional data for the others or removing that part.
- Line 42-43, What I understood this sentence to mean was diagnostics are important to protect public and veterinary health from the zoonotic potential of Brucella, but the phrase "prevent the spill over into the environment" is awkward and confusing. If wanting to include a comment about the environment, maybe rephrasing as protecting the environment or explaining where the spill over would be coming from would be helpful for clarification.
- Line 43, switch "is" to "are"
- Line 45, add a comma after laboratories and remove "one the"
- Line 48, remove "worldwide"
- Line 59, numbers are used and spelled out, use one form consistently (I.E. "12 Brucella species" and "three are considered...")
- Line 59-60 states 3 Brucella species are pathogenic to humans but does not list what these are. The species are listed in line 60 but discussed in the context of animal reservoirs. There needs to be a link or better clarification to these species as human pathogens.
- Line 61, add "respectively" after "pigs" and before "causing"
- Line 61-63, I believe you are trying to say that animals rarely show symptoms, but when clinical signs are displayed, they include what you have listed, but how it's currently phrased is a bit confusing and contradictory. I would also recommend including a descriptor infectious/non-infectious for the long-lasting vaginal discharge.
- Line 63, clarify "Excretion" more - I assume you are discussing excretion of the Brucella bacteria, but excretion could be construed as bodily excretions as well since discharges were discussed in the previous sentence.
- Line 66, change "get" to "become"
- Line 67, add "contaminated" between "with" and "animal" and replace "and" with "or a"
- Line 67, add a comma after "humans" and add "a" after usually

- Line 69, add "a" between "has" and "low"
- Line 70, change "abscess" to "abscesses"
- Line 71, clarify the phrase "tendency of the disease to present in a chronic...". Is this in people? In animals? In both? As it is, I'm assuming it is describing humans.
- Line 73, remove the comma after "miscarriage" but I would rearrange the sentence to start with the phrase "In animals, isolation can be done..." and move the "after miscarriage or other clinical manifestations" phrase to the end of the sentence.
- Line 74, EU and OIE are used and though they are common abbreviations, I would recommend spelling them out the first time they are used here.
- Line 75, add a comma after "years"
- Line 76, remove "more and more"
- Line 77, clarify "bacterial detection" method. Are you meaning culture or PCR or both?
- Line 78, replace "have" with "has" and end the sentence after "survival". Start a new sentence at "Freezing"
- Line 79, replace "as" with "is"
- Line 81, add a qualifier to "personnel" - is the personnel sampling? Or personnel testing? Or other?
- Line 82, replace "is" with "the" and "even more" with "especially"
- Line 85, "Referent" should this be "Reference"? also clarification on the "not more than 48 hours after sampling" phrase is needed. Is this for sample submission (I.E. when the sample is originally mailed) or for sample arrival at the lab?
- Line 88, "Which means that on average" is awkwardly phrased here. Using something more like "With this timeframe" or "With this submission timeline"
- Line 89, replace "the" with "a"
- Line 92, end the sentence after "detection", remove "whereas" and start the next sentence with "PCR..."
- Line 99, add a comma after "objectives"
- Line 100 discusses 2 storage conditions but does not mention timeframes which many comparisons are used later to the immediate testing. Maybe rephrase to include 2 timeframes and 2 temperature conditions?
- Line 110, "used compared" only need "used" or "compared" here
- Line 118, replace "dilutions" with "dilution" and "were" with "was"
- Line 122, it would be nice to add more information as to why this size of swab was selected, even if it is something as simple as this size is what is used for official testing, or commonly used in private practice, etc. Also, I'm sure more than 8 swabs were identified in review of commercially available swabs. Maybe some discussion as to how many were reviewed and how these 8 were identified for use (any criteria the others did not meet for inclusion, limiting to number by material type, etc.) should be included?
- Line 130-131, "until full capacity of the tip was reached" needs more clarification and discussion. How was this determined? Evidenced by dripping from the swab? What signaled the endpoint?
- Line 142, clarify the "two independent analyses" phrase. Did multiple individuals perform the spiking and testing? Was it the same individual doing multiple runs?
- Line 147, make "Swabs" singular and "tip" plural
- Line 147, clarify which swab tips were removed? Was this just for the eSwabTM or was it for all swabs. Since the previous sentence does not mention swab removal from the PBS clarification is needed.
- Line 151, replace "dilutions" with "dilution"
- Line 153, "at least in duplicate" needs revised. Does this mean the protocol was different for different swabs? How did data collection change if there were swabs with 2 plates and other swabs with more than 2 plates? If this was how it was done, there needs to be further explanation. If only 2 plates were used, this needs to be stated that way.
- Line 164, remove "Total" and replace "were" with "was"
- Line 172, add "the" between "to" and "appropriate"
- Line 198, remove the first "The"
- Line 199, replace the second "the" with "a"
- Line 203, percentage listed is mean recovery rate, but before is listed as mean survival rate, please keep consistent terminology
- Line 203-204, again recovery rate is discussed, same comment as above about consistent terminology
- Line 205, discusses recovery ratios - same comments as above
- Line 207, add "the" between "between" and "three"
- Line 208, would rephrase as "The bacteriology recovery of all three Brucella strains..." and provide the p-value to show the significance
- Line 217, give the percentages for the recovery rates from the different storage temperatures as done in the line before for immediately treated swabs
- Line 219, add "when" between "than" and "frozen"
- Line 219-221 is phrased awkwardly, maybe rephrase as "Live bacteria were isolated with at least 20% recovery from all directly treated swabs". Also, if you want to include the one swab type exception, explain which one it was any what differences there were. Also, I would give the percentages for the live bacteria recovery from the other storage conditions here for an easy comparison for the reader.
- Line 222-224 can be condensed by removing as the comparisons listed are by storage conditions and are redundant.
- Line 232, add a comma after "one"
- Line 233, replace the last "the" with "a"
- Line 237, replace the ";" with "and was" and add "the" between "from" and "control"

- Line 238, replace "none" with "no" and add "the" between "to" and "control"
- Line 239, remove "it"
- Line 232-239, add percentages for the recovery rates. This is done some in the next section discussing the qPCR and should be done for the live bacteria recovery as well. The results section should have numbers or percentages shown to back the statements.
- Line 242, add numbers or percentages to the comparison of dry, non-flocked swabs to controls.
- Line 243, give values for the comparison of S4 to S7
- Line 245, remove the comma after "isolation" and add "a" after "(S5)"
- Line 247-249, give numbers and/or percentages. Provide the data
- Line 254-262, same comment as above
- Line 264-265, move the "except S5" to the end of the sentence and give the numbers to show the exception
- Line 267, "swabs" should be singular
- Line 263-270, give numbers and/or percentages. Provide the data.
- Line 273, replace "sampled" with "are used for sampling"
- Line 274, "on to" should be one word
- Line 276, add a comma after "manual"
- Line 277, replace "and" with "or"
- Line 278-281 Sentence/thought should be expounded upon more. The fact that Brucella is excreted in the billions or trillions after miscarriages is important because...there's always excess bacteria? Why is the bacterial excretion fact important to transport information or lack thereof?
- Line 282, remove "a"
- Line 286, "...after 72h at +4oC or at -20oC." I would recommend re-writing to something like "after 72h at +4oC and after 72h at -20oC." As written, it seems like the swabs were +4oC or -20oC.
- Line 286, remove "most"
- Line 290, remove the comma after "swabs"
- Line 294, "rescued" is used instead of "recovered" as previously used. Please use consistent terminology
- Line 294, "sample" should be plural
- Line 295, add "over" between "However" and "the" and add a comma after "decade"
- Line 296, the phrase "and assured the constant conditions" is awkward. Recommend clarifying or rephrasing that part.
- Line 298, remove "significantly"
- Line 308-309, The use of refrigeration in a shorter timeframe should be addressed here. Have studies shown shorter times than 72 hours to be advantageous? Is this an area for further study?
- Line 315, remove the comma after "methods"
- Line 326, add "an" between "is" and "even"
- Line 329, remove "In the same time" and start the sentence at "The higher..."
- Line 332, replace "a" with "the" and add "the" between "than" and "others" and make "others" singular
- Line 336, "diagnostics" should be singular
- Line 337, remove "bacteria"
- Line 343, "fibbers" should be "fibers"
- Line 344, switch the order of "impacts" and "significantly"
- Line 347, "protect" should be "protects"
- Line 351, "within" should be "with"
- Line 352, "diagnostics" should be singular
- Line 350-353 should be expanded to discuss the fact that only 1 swab with protective media was tested and address the fact that additional testing of swabs with protective media should be performed to determine the best protective media and to confirm that repeatability to the results here
- Line 353, add "to" between "seems" and "play"
- Line 354, remove the comma after "DNA"
- Line 356, add "an" between "in" and "environment"
- Line 360, remove "a"
- Line 364, "swab" should be plural
- Line 367, add "compared to the" between "as" and "positive"
- Line 371, remove "as"
- Line 377, List "(S6)" after "Ca-alginate" as done previously throughout the paper
- Line 378, add "the" between "Although" and "Ca-alginate"
- Line 379, clarify "it" - referring to the swab or testing and remove "the"
- Line 380, remove "These" and start the sentence with "Swabs"
- Line 372-383, discussion of additional testing of the swabs at their lowest absorbance level and how that could change results should be included. Also, discussion of additional testing of these swabs using a matrix more like the discharges sampled when used in the field should be done.
- Line 392, remove "in neither"
- Line 393, add a comma after "conclusion"
- Line 394, remove "in the same time" and the "the" between "prevent" and "spill"
- Line 397, remove "present"

- Line 399, "swab" should be plural
- Line 423-424, check that this is cited correctly. 2016 and 2019 are listed, but the most recent version I have been able to find is 2018.
- Figure 1 and Figure 2 confidence intervals are listed in the legends with the asterisks, but how they are used in the figures is very confusing. The CI is reported between groups, but this is not clear in the figures. Possibly using different symbols for each comparison could help.
- Line 557, remove "various"
- Line 562-563, remove "regardless the strain" as this figure only shows the relationship between live bacteria/DNA and temperature condition
- Line 563, "Full and dashed lines..." please add this to the legend for Figure 1.
- Line 569, "PC" is used as an abbreviation for positive control, but it is "CP" in the figures. Please use abbreviations consistently
- Line 571, "depend" should be "depended"
- Line 572, "show" should be "showed the"
- Line 575, remove the comma after "results"
- Line 576, add a comma after "Still" and "release" should be "released" and "is" should be "was"
- Line 578, "negatively" and "affected" should have their order reversed in the sentence
- Line 591, the ";" after "S2" should be a comma, remove "even", end the sentence after "isolated", and remove "while only from" and add "were the only cases where"
- Line 592, remove the second "from"
- Line 593, add "had" between "flocked)" and "significantly", also "significantly" is used quite a lot here, maybe find another word/phrase to use so it's not as repetitive.
- Line 592-593, sentence needs a little revision and more bacteria were isolated from S5 compared to what? Clarify in the sentence
- Line 593 remove "From" and start the sentence with the swab types then add "had" before "significantly"
- Line 594, remove the first "were"
- Line 594-596, it makes it seem like S5 was the only swab to have growth when others did not. Is this what you were trying to say? Please revise for clarification if it is not.
- Line 596, move "Throughout the storage conditions" to the end of the sentence.
- Line 601, "this variation was less significant when they were frozen at..." is the -20oC compared to the +4oC or to the positive control or to the DT? Please clarify.
- Line 604, CI were listed out in previous figures where this only shows one CI, can you show the range as the previous figures with different symbols?
- Line 613, remove "the amplification of IPC"
- Line 619, "difference" should be "different"
- Line 620, replace "with" with "compared to the"

Staff Comments:

Preparing Revision Guidelines

For complete guidelines on revision requirements, please see the Instructions to Authors at [link to page]. **Submissions of a paper that does not conform to Microbiology Spectrum guidelines will delay acceptance of your manuscript.**

Please return the manuscript within 60 days; if you cannot complete the modification within this time period, please contact me. If you do not wish to modify the manuscript and prefer to submit it to another journal, please notify me of your decision immediately so that the manuscript may be formally withdrawn from consideration by Microbiology Spectrum.

If you would like to submit an image for consideration as the Featured Image for an issue, please contact Spectrum staff.

If your manuscript is accepted for publication, you will be contacted separately about payment when the proofs are issued;

please follow the instructions in that e-mail. Arrangements for payment must be made before your article is published. For a complete list of **Publication Fees**, including supplemental material costs, please visit our website.

In this study, Freddi et al examine the impact of different commercially available swabs on Brucella culture and PCR detection. It highlights the importance of collection media and storage conditions in diagnostics. My comments are as follows:

Major comments:

1. The entire manuscript needs to proofread for grammar and sentence structures, especially the Abstract, Importance, and Introduction sections
2. The swabs were assessed by inoculating them with $\sim 10^6$ Cfu/50 uL of bacteria. Why was this the appropriate Cfu to assess these swabs?
3. The authors conclude that the flocked swabs with protective media enhance the recovery of Brucella in culture and for molecular detection. This study would be of a greater impact if the authors could collect some prospective data from the field to support their observations.

Minor comments

None

Dear Dr Lainhart,

The authors are grateful to the reviewers for their critical evaluation and their helpful suggestions and corrections. We have made changes according to their constructive comments and introduced modifications to the manuscript to clarify our work. All issues raised by the reviewers have been addressed here below. The corrections are visible in the revised manuscript (deleted are marked with yellow and added text in blue).

If you consider that additional changes would be needed we rest at your disposal.

All the best!

Luca Freddi

Reviewer #1 (Comments for the Author):

In this study, Freddi et al examine the impact of different commercially available swabs on *Brucella* culture and PCR detection. It highlights the importance of collection media and storage conditions in diagnostics. My comments are as follows:

Major comments:

1. The entire manuscript needs to proofread for grammar and sentence structures, especially the Abstract, Importance, and Introduction sections

All grammar and wording mistakes have been checked and revised in the new version.

2. The swabs were assessed by inoculating them with $\sim 10^6$ Cfu/50 uL of bacteria. Why was this the appropriate Cfu to assess these swabs?

Because it is 10^6 CFU/50 μ l is enough quantity of bacteria to ensure positive detection (by qPCR and bacterial culture) for all types of swabs immediately processed after spiking. Non-published preliminary tests identify $\sim 10^6$ CFU/50 μ l of bacteria as the minimal concentration for spiking swabs. In the same time, it is a small enough quantity to have negative values if the swabs do not retain *Brucellae* or their DNA. In authors experience with field samples, when contaminated with 10^6 CFU/50 μ l the isolation rate is above 95%. As show in Fig. 2A, only from a very small number of swabs positive *Brucella* culture cannot be ensured, after directly treatment.

3. The authors conclude that the flocked swabs with protective media enhance the recovery of *Brucella* in culture and for molecular detection. This study would be of a greater impact if the authors could collect some prospective data from the field to support their observations.

The aim of this study was to compare the different swab types in the as equal as possible conditions. Therefore, laboratory conditions, where each swab could be charged with the same quantity of *Brucellae* is the ideal for selecting the ones that can be used in the field conditions.

According to the reviewer's comments, the importance to compare selected swabs in the field conditions is highly interesting and will be focus on soon.

Minor comments

None

Reviewer #2 (Comments for the Author):

Overall Comments:

- The paper was an interesting read and I appreciated the opportunity to review it!

We would like to thank the reviewer for this positive evaluation.

- Overall, it is well done, but there are a few areas/phrases that seem awkward that I would recommend changing for clarity (described in detail below).

All grammar and wording mistakes have been checked and revised in the new version.

- It would be nice to have a little more discussion on why the 3 *Brucella* strains were selected somewhere in the paper. You give a brief glimpse into why the Thomsen strain was selected with the discussion about its lower resistance to storage, but more discussion about the other strains would be appreciated.

According to reviewer remark, the text has been modified:

Line 62 – Sentence added: These three species of *Brucella*, are mandatory reportable diseases throughout the world, and still have high prevalence among domestic and wild animals, causing threat for public health (11).

Line 366 – Sentence added: From our experience, besides lower detectability in non- and selective media, Thomsen strain grows slower and takes several days to reach visible CFUs.

- The positive and negative controls are discussed in the experimental design section, but only discussed so far as what they are. Since a lot of your experiment and data is comparison to the positive control, I would also recommend detailing how the positive and negative controls were tested. Was it placed on a control swab and tested as well? Was the PBS solution directly tested? Were the controls tested in quadruplicate as the swabs were? Clarification and more discussion on this would be useful.

We apologise, authors considered that positive and negative controls should be tested in the same way as analysed materials. To correct that from line 137 several additions were made: The same amount of bacteria was also used to spike 2 ml of PBS as a positive control (PC) (50°µl of bacteria + 1950°µl of PBS), and a 2 ml non-spiked PBS was used as a negative control. Three storage conditions were selected to compare swabs, PCs and negative controls:

immediately processed after spiking (DT); stored for three days at +4°C (72h_+4°C) and -20°C (72h_-20°C). For all storage conditions, of each *Brucella* strain used, four swabs per type as well as four PCs were spiked. In order to avoid the biological effect of swab material and variations of bacteria pipetting, two independent analyses were performed for each individual swab, accounting in total of eight analyses per swab type, bacterial strain, and storage condition. To compare culture and molecular analysis, spiked swab tips were immersed in 2 ml of PBS, vortexed for 30 seconds and incubated at 37°C overnight, to discharge bacteria. Only to the eSwab™ 1 ml of PBS was added because it already contains 1 ml of transport medium. PCs were only incubated at 37°C overnight, without additional PBS.

- Along those lines, the data shows that one swab type (S5) produced more live bacteria and DNA when compared to the positive control, and is pointed out several times, but there is no discussion as to how this would be possible. This should be something to clarify and discuss further.

We agree with the reviewer regarding the importance to understand how the eSwab™ (S5) can provide better bacterial and DNA protection than other treated swabs. This topic was discussed extensively in the discussion session (lines 349-368). The swab S5 has the modified Amies medium that plays important role in protecting bacteria and DNA. In particular, the presence of the emulsifier in the medium can explain a better preservation of live bacteria and DNA over time. Moreover, the flocked swab (as swab S4) contain a nylon flocked fiber which can increase from two- to three-fold the absorption of yield targeted cell and improve diagnostic sensitivity.

- I would like to see more data from your testing. I would recommend including a table of results showing the calculated percent recovery of live *Brucella* and the calculated percent recovery of *Brucella* DNA. Also, it would be nice to visually see the correlation of Ct values and number of CFU's for each *Brucella* species.

According to reviewer remark, authors provide the mean, minimal and maximal values of bacterial and DNA recovery rates regarding the *Brucella* strains, storage conditions, and swab type used in additional supplementary Table S2. Moreover authors are ready and willing (as we stated) to share the raw data upon individual request for future studies or analyses.

- More numbers and data need to be given in the Results section. The tables and data shown are more comparisons of the values, but not the values themselves so there is nothing for the reader to assess except your statements about the data.

The results section has been revised to add data (where possible) on mean values for bacteriology and qPCR recovery rate. For data not shown, authors provide the mean, minimal and maximal values of bacterial and DNA recovery rates regarding the *Brucella* strains, storage conditions, and swab type used in additional supplementary table S2.

- Was any method or corrections used in the statistical analysis for repeat testing as the swabs/testing was performed in quadruplicate? This needs to be addressed.

There was no need for corrections. As S1 table indicates, the IPC was always in the acceptable range of 3Ct differences (the maximal difference observed was 2Ct). Therefore, the authors considered that there is no need for correction in live bacteria recovery, because of the discrepancies in results.

- Daub and swab seem to be used interchangeably throughout the paper but can be confusing. I would recommend only using a single terminology consistently.

Daub and swab are the nouns for the same objects, and are used interchangeably in order to avoid repetitions, make the sentences wordy and confusing. As manuscript has been proofread, we removed word daub wherever was possible.

- Use of duplicate/triplicate/quadruplicate should be singular, not plural as stated multiple times within the paper.

Corrected.

Specific Comments:

- Line 24, remove "both" as you list more than 2 areas where Brucella can be persistent

Done

- Line 26, "Therefore" seems to be an awkward transition phrase here. Recommend re-writing or adding another sentence to connect the thoughts from this sentence to the one before it.

Corrected to: Swabs are used for sampling in bacteriological and/or molecular diagnostics, from seropositive animals with disease symptoms, from genitalia or tissue lesions as well as from contaminated environment.

- Line 27, remove "the"

Done

- Line 29-30, "...optimal storage conditions for shipment within 72 hours." I believe you mean to say optimal storage conditions and timeframe for testing. The way this is phrased is awkward and seems to mean anytime within those 72 hours, but your experiments were either immediate or 3 days with no in-between.

According to reviewer remark, the text has been corrected to: and determine the optimal storage conditions and time frame for testing.

- Line 32, "...after 72h at +4oC or at -20oC." I would recommend re-writing to something like "after 72h at +4oC and after 72h at -20oC." As written, it seems like the swabs were +4oC or -20oC.

The text has been corrected following reviewer suggestion.

- Line 32, remove "That"

Corrected.

- Line 34, make "swabs" singular as there was only 1 tested with protective media

Corrected.

- Line 35, 55% is listed but there's no reference to the reader on how that compares to the other swabs to know if 55% is good or bad. I'd recommend adding additional data for the others or removing that part.

According to reviewer remark, the text has been corrected to: The flocked swab immersed in protective media, provides the best conditions for *Brucella* survival in all three storage conditions.

- Line 42-43, What I understood this sentence to mean was diagnostics are important to protect public and veterinary health from the zoonotic potential of *Brucella*, but the phrase "prevent the spill over into the environment" is awkward and confusing. If wanting to include a comment about the environment, maybe rephrasing as protecting the environment or explaining where the spill over would be coming from would be helpful for clarification.

According to reviewer remark, the text has been corrected to: In order to protect public and veterinary health from highly zoonotic bacteria such as *Brucella* genus, and prevent the dissemination into the environment, the direct diagnostics are of utmost importance.

- Line 43, switch "is" to "are"

Corrected.

- Line 45, add a comma after laboratories and remove "one the"

Corrected.

- Line 48, remove "worldwide"

Corrected.

- Line 59, numbers are used and spelled out, use one form consistently (I.E. "12 *Brucella* species" and "three are considered...")

According to standard style guides, the authors use words for cardinal numbers less than 10 and numerals for 10 and above.

- Line 59-60 states 3 *Brucella* species are pathogenic to humans but does not list what these are.

The species are listed in line 60 but discussed in the context of animal reservoirs. There needs to be a link or better clarification to these species as human pathogens.

According to reviewer remark, the text has been corrected to: Out of the 12 currently described *Brucella* species, *B. abortus*, *B. melitensis* and *B. suis* are considered to be highly pathogenic for humans (4, 5). These three species of *Brucella*, are mandatory reportable diseases throughout the world, and still have high prevalence among domestic and wild animals, causing threat for public health. They affect mainly cattle, small ruminants and pigs respectively, causing abortions and infertility (6, 7).

- Line 61, add "respectively" after "pigs" and before "causing"

Corrected.

- Line 61-63, I believe you are trying to say that animals rarely show symptoms, but when clinical signs are displayed, they include what you have listed, but how it's currently phrased is a bit confusing and contradictory. I would also recommend including a descriptor infectious/non-infectious for the long-lasting vaginal discharge.

The text has been modified to: Infected animals rarely show clinical symptoms. Those usually occur during pregnancy in females, as miscarriages which can also be followed by long-lasting vaginal discharge and in worst case end-up in sterility (6).

- Line 63, clarify "Excretion" more - I assume you are discussing excretion of the *Brucella* bacteria, but excretion could be construed as bodily excretions as well since discharges were discussed in the previous sentence.

The text has been modified to: Excretion of *Brucella* is reported in sick as well as asymptomatic animals.

- Line 66, change "get" to "become"

Corrected.

- Line 67, add "contaminated" between "with" and "animal" and replace "and" with "or a"

Corrected.

- Line 67, add a comma after "humans" and add "a" after usually

Corrected.

- Line 69, add "a" between "has" and "low"

Corrected.

- Line 70, change "abscess" to "abscesses"

Corrected.

- Line 71, clarify the phrase "tendency of the disease to present in a chronic...". Is this in people? In animals? In both? As it is, I'm assuming it is describing humans.

According to reviewer remark, the text has been modified: The tendency of the disease to present a chronic and persistent form in humans, results in granulomatous disease capable of affecting any organ system (1).

- Line 73, remove the comma after "miscarriage" but I would rearrange the sentence to start with the phrase "In animals, isolation can be Corrected..." and move the "after miscarriage or other clinical manifestations" phrase to the end of the sentence.

According to reviewer remark, the text has been modified: In animals, isolation of *Brucellae* can be done via collection of genital swabs after miscarriage or other clinical manifestations.

- Line 74, EU and OIE are used and though they are common abbreviations, I would recommend spelling them out the first time they are used here.

Corrected.

- Line 75, add a comma after "years"

Corrected.

- Line 76, remove "more and more"

Corrected.

- Line 77, clarify "bacterial detection" method. Are you meaning culture or PCR or both?

According to reviewer remark, the text has been modified to: Swab type and storage temperature could influence successful bacterial isolation and molecular detection

- Line 78, replace "have" with "has" and end the sentence after "survival". Start a new sentence at "Freezing"

The sentence has been modified following reviewer's suggestion.

- Line 79, replace "as" with "is"

Corrected.

- Line 81, add a qualifier to "personnel" - is the personnel sampling? Or personnel testing? Or other?

The sentence has been completed following reviewer's suggestion.

- Line 82, replace "is" with "the" and "even more" with "especially"

Corrected.

- Line 85, "Referent" should this be "Reference"? also clarification on the "not more than 48 hours after sampling" phrase is needed. Is this for sample submission (I.E. when the sample is originally mailed) or for sample arrival at the lab?

According to reviewer remark, the text has been revised to: OIE guidelines instruct that field samples should be delivered to the diagnostic laboratories as soon as possible (11), while the EU Reference Laboratory for Brucellosis recommends a maximum delay of 48 hours between sampling and its arrival to the laboratory...

- Line 88, "Which means that on average" is awkwardly phrased here. Using something more like "With this timeframe" or "With this submission timeline"

According to reviewer remark, the text has been modified to: With this submission timeline, diagnostics are performed, on average, three days after sample collection.

- Line 89, replace "the" with "a"

Corrected.

- Line 92, end the sentence after "detection", remove "whereas" and start the next sentence with "PCR..."

Corrected.

- Line 99, add a comma after "objectives"

Corrected.

- Line 100 discusses 2 storage conditions but does not mention timeframes which many comparisons are used later to the immediate testing. Maybe rephrase to include 2 timeframes and 2 temperature conditions?

According to reviewer remark, the text has been revised to: In order to achieve the objectives, we tested bacterial culture and qPCR diagnostics on nine different spiked swabs, treated immediately or after 72h and stored in two temperature conditions.

- Line 110, "used compared" only need "used" or "compared" here

Corrected.

- Line 118, replace "dilutions" with "dilution" and "were" with "was"

Corrected.

- Line 122, it would be nice to add more information as to why this size of swab was selected, even if it is something as simple as this size is what is used for official testing, or commonly used in private practice, etc. Also, I'm sure more than 8 swabs were identified in review of commercially available swabs. Maybe some discussion as to how many were reviewed and how these 8 were identified for use (any criteria the others did not meet for inclusion, limiting to number by material type, etc.) should be included?

According to reviewer remark, the text has been modified to: Among commercially-available swabs, only swabs with regular sized tips (~5 mm wide and ~16 mm long) commonly used for sampling, were selected. Only swabs with a transport tube have been selected, without considering the different types of handles. Furthermore, for the same type of tip material, only one representative was selected, belonging to the main brands on the market.

- Line 130-131, "until full capacity of the tip was reached" needs more clarification and discussion. How was this determined? Evidenced by dripping from the swab? What signaled the endpoint?

According to reviewer remark, the text has been revised to: The absorbance of each swab type was calculated by adding 5 μ l of PBS until full capacity of the tip was reached, evidenced by dripping from the swab.

- Line 142, clarify the "two independent analyses" phrase. Did multiple individuals perform the spiking and testing? Was it the same individual doing multiple runs?

According to reviewer remark, the text has been modified to: In order to avoid the biological effect of swab material and variations of bacteria pipetting, two independent analyses were performed by the same person for each individual swab, accounting in total of eight analyses per swab type, bacterial strain, and storage condition.

- Line 147, make "Swabs" singular and "tip" plural

Corrected.

- Line 147, clarify which swab tips were removed? Was this just for the eSwabTM or was it for all swabs. Since the previous sentence does not mention swab removal from the PBS clarification is needed.

According to reviewer remark, the text has been modified to: All swab tips were then removed after vortexing step and the Working Suspension (WS) was used for bacterial plate culturing and DNA extraction.

- Line 151, replace "dilutions" with "dilution"

Corrected.

- Line 153, "at least in duplicate" needs revised. Does this mean the protocol was different for different swabs? How did data collection change if there were swabs with 2 plates and other swabs with more than 2 plates? If this was how it was done, there needs to be further explanation. If only 2 plates were used, this needs to be stated that way.

Corrected to: All dilutions were inoculated in duplicates.

- Line 164, remove "Total" and replace "were" with "was"

Corrected.

- Line 172, add "the" between "to" and "appropriate"

Corrected.

- Line 198, remove the first "The"

Corrected.

- Line 199, replace the second "the" with "a"

Corrected.

- Line 203, percentage listed is mean recovery rate, but before is listed as mean survival rate, please keep consistent terminology

Corrected.

- Line 203-204, again recovery rate is discussed, same comment as above about consistent terminology

Corrected.

- Line 205, discusses recovery ratios - same comments as above

Corrected.

- Line 207, add "the" between "between" and "three"

Corrected.

- Line 208, would rephrase as "The bacteriology recovery of all three *Brucella* strains..." and provide the p-value to show the significance

According to reviewer remark, the text has been revised to: The bacteriology recovery of all three *Brucella* strains (mean values 55.8%, 50.5% and 37.7%, for 16M, 544 and Thomsen respectively) was significantly lower than molecular detection (mean values 66.9%, 67.2% and 69.1%, respectively; p -values < 0.00001, compared to both conditions), especially for Thomsen (data not shown).

- Line 217, give the percentages for the recovery rates from the different storage temperatures as done in the line before for immediately treated swabs

According to reviewer remark, the text has been modified to: The recovery rate of immediately treated swabs (mean value 68.2%) is significantly higher than recovery from daubs stored at +4°C or -20°C (mean values 48.2% and 27.5% respectively; p -values < 0.00001, compared to both conditions).

- Line 219, add "when" between "than" and "frozen"

Corrected.

- Line 219-221 is phrased awkwardly, maybe rephrase as "Live bacteria were isolated with at least 20% recovery from all directly treated swabs". Also, if you want to include the one swab type exception, explain which one it was any what differences there were. Also, I would give the percentages for the live bacteria recovery from the other storage conditions here for an easy comparison for the reader.

The paragraph has been revised following reviewer's suggestion.

- Line 222-224 can be condensed by removing as the comparisons listed are by storage conditions and are redundant.

Because of the amount of comparisons presented, the authors feel the additional clarification, and somewhat repetition is useful in this paragraph.

- Line 232, add a comma after "one"

Corrected.

- Line 233, replace the last "the" with "a"

Corrected.

- Line 237, replace the ";" with "and was" and add "the" between "from" and "control"

Corrected.

- Line 238, replace "none" with "no" and add "the" between "to" and "control"

Corrected.

- Line 239, remove "it"

Corrected.

- Line 232-239, add percentages for the recovery rates. This is done some in the next section discussing the qPCR and should be done for the live bacteria recovery as well. The results section should have numbers or percentages shown to back the statements.

The paragraph has been revised following reviewer's suggestion to include result section.

- Line 242, add numbers or percentages to the comparison of dry, non-flocked swabs to controls.

The data were shown at the top of the paragraph.

- Line 243, give values for the comparison of S4 to S7

The data were shown at the top of the paragraph.

- Line 245, remove the comma after "isolation" and add "a" after "(S5)"

Corrected.

- Line 247-249, give numbers and/or percentages. Provide the data

The paragraph has been revised following reviewer's suggestion to include result section.

- Line 254-262, same comment as above

The paragraph has been revised following reviewer's suggestion to include result section.

- Line 264-265, move the "except S5" to the end of the sentence and give the numbers to show the exception

Corrected.

- Line 267, "swabs" should be singular

Corrected.

- Line 263-270, give numbers and/or percentages. Provide the data.

The paragraph has been revised following reviewer's suggestion to include result section.

- Line 273, replace "sampled" with "are used for sampling"

Corrected.

- Line 274, "on to" should be one word

Corrected.

- Line 276, add a comma after "manual"

Corrected.

- Line 277, replace "and" with "or"

Corrected.

- Line 278-281 Sentence/thought should be expounded upon more. The fact that *Brucella* is excreted in the billions or trillions after miscarriages is important because...there's always excess bacteria? Why is the bacterial excretion fact important to transport information or lack thereof?

According to reviewer suggestion, the text has been revised to: This lack of information may be overcome by the fact that when there is a suspicion on *Brucellae* as causative agents of miscarriage, bacteria can be excreted in billions or trillions (up to 10^{13} bacteria) for a short period and potentially easily detected with all type of swabs (8). However, when the animal is serologically positive, but without specific clinical manifestations (fever, cutaneous wounds, orchitis, epididymitis, lymphadenitis, etc.) or in environmental swabs, the number of suspected bacteria could be far lower making detection more challenging.

- Line 282, remove "a"

Corrected.

- Line 286, "...after 72h at +4oC or at -20oC." I would recommend re-writing to something like "after 72h at +4oC and after 72h at -20oC." As written, it seems like the swabs were +4oC or -20oC.

Corrected.

- Line 286, remove "most"

Corrected.

- Line 290, remove the comma after "swabs"

Corrected.

- Line 294, "rescued" is used instead of "recovered" as previously used. Please use consistent terminology

Corrected.

- Line 294, "sample" should be plural

Corrected.

- Line 295, add "over" between "However" and "the" and add a comma after "decade"

Corrected.

- Line 296, the phrase "and assured the constant conditions" is awkward. Recommend clarifying or rephrasing that part.

According to reviewer suggestion, the text has been revised to: However, over the last decade, advancements in sample preservation resulted in the improved transport assuring constant conditions and their traceability (as temperature, humidity, time, etc...).

- Line 298, remove "significantly"

Corrected.

- Line 308-309, The use of refrigeration in a shorter timeframe should be addressed here. Have studies shown shorter times than 72 hours to be advantageous? Is this an area for further study?

The authors found no comprehensive study in the literature demonstrating an advantage to freeze swabs for a short period, less than 72h. The few non-exhaustive studies, seem to show that freezing at -20 ° C for a period between 24-48h compared to storage at room temperature, +5°C and -80°C, does not bring significant advantages for the recovery of total DNA (20) or bacterial viability (Yagüe et. al 2021, Journal of Applied Microbiology: <https://doi.org/10.1111/jam.15023>).

- Line 315, remove the comma after "methods"

Corrected.

- Line 326, add "an" between "is" and "even"

Corrected.

- Line 329, remove "In the same time" and start the sentence at "The higher..."

Corrected.

- Line 332, replace "a" with "the" and add "the" between "than" and "others" and make "others" singular

Corrected.

- Line 336, "diagnostics" should be singular

Corrected.

- Line 337, remove "bacteria"

Corrected.

- Line 343, "fibbers" should be "fibers"

Corrected.

- Line 344, switch the order of "impacts" and "significantly"

Corrected.

- Line 347, "protect" should be "protects"

Corrected.

- Line 351, "within" should be "with"

Corrected.

- Line 352, "diagnostics" should be singular

Corrected.

- Line 350-353 should be expanded to discuss the fact that only 1 swab with protective media was tested and address the fact that additional testing of swabs with protective media should be performed to determine the best protective media and to confirm that repeatability to the results here

According to reviewer suggestion, the sentence has been added at line 391: In our experiments only one type of swab with protective media has been tested, and further studies should be performed to compare the usefulness of other available media and bacterial preservatives

- Line 353, add "to" between "seems" and "play"

Corrected.

- Line 354, remove the comma after "DNA"

Corrected.

- Line 356, add "an" between "in" and "environment"

Corrected.

- Line 360, remove "a"

Corrected.

- Line 364, "swab" should be plural

Corrected.

- Line 367, add "compared to the" between "as" and "positive"

Corrected.

- Line 371, remove "as"

Corrected.

- Line 377, List "(S6)" after "Ca-alginate" as done previously throughout the paper

Corrected.

- Line 378, add "the" between "Although" and "Ca-alginate"

Corrected.

- Line 379, clarify "it" - referring to the swab or testing and remove "the"

Corrected.

- Line 380, remove "These" and start the sentence with "Swabs"

Corrected.

- Line 372-383, discussion of additional testing of the swabs at their lowest absorbance level and how that could change results should be included. Also, discussion of additional testing of these swabs using a matrix more like the discharges sampled when used in the field should be included.

According to reviewer remark, the sentence has been added: This data should be finally compared with matrices usually collected in the field conditions, in order to estimate swabs' absorption variations.

Line 420: It would also be interesting, to identify which swabs provide the best bacterial survival and DNA retention at their lowest absorbance level, mimicking situation with limited amount of sample.

- Line 392, remove "in neither"

Corrected.

- Line 393, add a comma after "conclusion"

Corrected.

- Line 394, remove "in the same time" and the "the" between "prevent" and "spill"

Corrected.

- Line 397, remove "present"

Corrected.

- Line 399, "swab" should be plural

Corrected.

- Line 423-424, check that this is cited correctly. 2016 and 2019 are listed, but the most recent version I have been able to find is 2018.

Corrected.

- Figure 1 and Figure 2 confidence intervals are listed in the legends with the asterisks, but how they are used in the figures is very confusing. The CI is reported between groups, but this is not clear in the figures. Possibly using different symbols for each comparison could help.

According to reviewer remark, Figures 1 and 2 have been modified to make more visible which comparison the asterisks refer to. The number of abstracts expresses the p-value between the groups compared and marked by the continuous line.

- Line 557, remove "various"

Corrected.

- Line 562-563, remove "regardless the strain" as this figure only shows the relationship between live bacteria/DNA and temperature condition

Corrected.

- Line 563, "Full and dashed lines..." please add this to the legend for Figure 1.

Corrected.

- Line 569, "PC" is used as an abbreviation for positive control, but it is "CP" in the figures. Please use abbreviations consistently

The abbreviation "PC" was modified in Fig. 3 and 4.

- Line 571, "depend" should be "depended"

Corrected.

- Line 572, "show" should be "showed the"

Corrected.

- Line 575, remove the comma after "results"

Corrected.

- Line 576, add a comma after "Still" and "release" should be "released" and "is" should be "was"

Corrected.

- Line 578, "negatively" and "affected" should have their order reversed in the sentence

As standard, authors decided to show always more prominent positive and then negative changes.

- Line 591, the ";" after "S2" should be a comma, remove "even", end the sentence after "isolated", and remove "while only from" and add "were the only cases where"

According to reviewer suggestion, the text has been revised to: When swabs were directly treated (DT), from four (S1, S2, S6 and S7), significantly less bacteria were detected than from PC, while only from S4 and S5 more bacteria was isolated than control. In some cases from S2, S6 and S7 no CFUs were isolated.

- Line 592, remove the second "from"

Corrected.

- Line 593, add "had" between "flocked)" and "significantly", also "significantly" is used quite a lot here, maybe find another word/phrase to use so it's not as repetitive.

Corrected with the following two comments

- Line 592-593, sentence needs a little revision and more bacteria were isolated from S5 compared to what? Clarify in the sentence

Corrected.

- Line 593 remove "From" and start the sentence with the swab types then add "had" before "significantly"

According to reviewer suggestion, the text has been revised to: When swabs were stored for 72h at +4°C, only the use of S5 (wet flocked) led to isolation of considerably more bacteria.

- Line 594, remove the first "were"

Corrected.

- Line 594-596, it makes it seem like S5 was the only swab to have growth when others did not. Is this what you were trying to say? Please revise for clarification if it is not.

According to reviewer suggestion, the text has been revised to: When swabs were stored for 72h at -20°C, only from S5 the live bacteria were isolated in all spiked swab, regardless the *Brucella* strain used.

- Line 596, move "Throughout the storage conditions" to the end of the sentence.

Corrected.

- Line 601, "this variation was less significant when they were frozen at..." is the -20oC compared to the +4oC or to the positive control or to the DT? Please clarify.

According to reviewer suggestion, the text has been modified to: While the qPCR detection varied with swab type when they were DT and stored at +4°C compared to PC, this variation was less significant when they were frozen at -20°C for three days.

- Line 604, CI were listed out in previous figures where this only shows one CI, can you show the range as the previous figures with different symbols?

According to reviewer suggestion, the figure 4 has been revised in order to standardize display of the CI.

- Line 613, remove "the amplification of IPC"

Corrected.

- Line 619, "difference" should be "different"

Corrected.

- Line 620, replace "with" with "compared to the"

Corrected.

Line 669 – legend for the Table S2 added: Comparison of bacteriology and qPCR recovery rates based on different *Brucella* strains, storage conditions and swab types. The mean, minimal and maximal values of bacterial and DNA recovery rates are presented regarding the *Brucella* strains, storage conditions, and swab type used. The results show significant variations in recovery rates of live bacteria between various swabs, based on different *Brucella* strains and storage conditions. In the same time, the variations in detectability of *Brucella* DNA by qPCR are significantly lower than live bacteria recovery as three values (mean, min, max) are compared.

Staff Comments:

Preparing Revision Guidelines

- Point-by-point responses to the issues raised by the reviewers in a file named "Response to Reviewers," NOT IN YOUR COVER LETTER.
- Upload a compare copy of the manuscript (without figures) as a "Marked-Up Manuscript" file.

- Each figure must be uploaded as a separate file, and any multipanel figures must be assembled into one file.
- Manuscript: A .DOC version of the revised manuscript
- Figures: Editable, high-resolution, individual figure files are required at revision, TIFF or EPS files are preferred

For complete guidelines on revision requirements, please see the Instructions to Authors at [link to page]. **Submissions of a paper that does not conform to Microbiology Spectrum guidelines will delay acceptance of your manuscript.**

September 30, 2021

Dr. Luca Freddi
ANSES
Animal Health Laboratory
14 Rue Pierre et Marie Curie
Maisons-Alfort 94700
France

Re: Spectrum00728-21R1 (The use of flocked swab with protective media increases the recovery of live *Brucella* spp. and DNA detection)

Dear Dr. Luca Freddi:

Your manuscript has been accepted, and I am forwarding it to the ASM Journals Department for publication. You will be notified when your proofs are ready to be viewed.

Sincerely,

William Lainhart
Editor, Microbiology Spectrum
